# Bioelectrical Impedance Analysis in Patients Undergoing Major Head and Neck Surgery: A Prospective Observational Pilot Study

**DOI:** 10.3390/jcm10030539

**Published:** 2021-02-02

**Authors:** Sharon Tzelnick, Pierre Singer, Yoni Shopen, Limor Moshkovitz, Shlomo Fireman, Thomas Shpitzer, Aviram Mizrachi, Gideon Bachar

**Affiliations:** 1Department of Otorhinolaryngology Head and Neck Surgery, Rabin Medical Center—Beilinson Hospital, Petach Tikva 49100, Israel; yoni86@gmail.com (Y.S.); Thomas.shpitzer@gmail.com (T.S.); aviramguy@hotmail.com (A.M.); gidybahar@gmail.com (G.B.); 2Sackler Faculty of Medicine, Tel Aviv University, Tel Aviv 39040, Israel; pierre.singer@gmail.com (P.S.); dr.lipius@yahoo.com (S.F.); 3Department of General Intensive Care and Institute for Nutrition Research, Rabin Medical Center—Beilinson Hospital, Petach Tikva 49100, Israel; 4Department of Nutrition, Rabin Medical Center—Beilinson Hospital, Petach Tikva 49100, Israel; limor.moshkovitz@gmail.com; 5Department of Anesthesiology Rabin Medical Center—Beilinson Hospital, Petach Tikva 49100, Israel

**Keywords:** head and neck surgery, bioelectrical impedance analysis, perioperative complications

## Abstract

Background: Head and neck patients are prone to malnutrition. Perioperative fluids administration in this patient group may influence nutritional status. We aimed to investigate perioperative changes in patients undergoing major head and neck surgery and to examine the impact of perioperative fluid administration on body composition and metabolic changes using bioelectrical impedance. Furthermore, we sought to correlate these metabolic changes with postoperative complication rate. In this prospective observational pilot study, bioelectrical impedance analysis (BIA) was performed preoperatively and on postoperative days (POD) 2 and 10 on patients who underwent major head and neck surgeries. BIA was completed in 34/37 patients; mean total intraoperative and post-anesthesia fluid administration was 3682 ± 1910 mL and 1802 ± 1466 mL, respectively. Total perioperative fluid administration was associated with postoperative high extra-cellular water percentages (*p* = 0.038) and a low phase-angle score (*p* < 0.005), which indicates low nutritional status. Patients with phase angle below the 5th percentile at POD 2 had higher local complication rates (*p* = 0.035) and longer hospital length of stay (LOS) (*p* = 0.029). Multivariate analysis failed to demonstrate that high-volume fluid administration and phase angle are independent factors for postoperative complications. High-volume perioperative fluids administration impacts postoperative nutritional status with fluid shift toward the extra-cellular space and is associated with factors that increase the risk of postoperative complications and longer LOS. An adjusted, low-volume perioperative fluid regimen should be considered in patients with comorbidities in order to minimize postoperative morbidity.

## 1. Introduction

Major head and neck surgeries are associated with significant morbidity, especially in the perioperative period. In head and neck oncologic patients, impaired wound healing, high postoperative infection rates, and prolonged hospital stay are associated with poor preoperative nutritional status [1,2]. Malnutrition in these patients can result from mechanical obstruction and debilitating pain that may cause dysphagia, which, together with catabolic factors caused by the tumor, lead to impaired nutritional state and cancer cachexia [3,4,5]. Several nutritional parameters, such as biochemical (albumin, pre-albumin) and body mass index (BMI), have been investigated in order to predict postoperative outcomes in these patients. However, the data are inconsistent regarding their utility [6,7,8,9,10,11].

Recently, several studies have shown an association between perioperative high-volume fluid administration and postoperative complication rates in patients undergoing head and neck microvascular reconstruction [12,13].

Nutrition can also be expressed as body composition. In the last two decades, bioelectrical impedance analysis (BIA) has been used to estimate body composition by measuring body component resistance (R) and reactance (Xc) by recording a voltage drop in applied current. Resistance related to the amount of water present in the tissues and reactance causes the current to lag behind the voltage, creating a phase shift [14]. One of the most clinically established impedance parameters in BIA is the phase angle. Phase angle, a marker of soft-tissue mass and hydration status, is positively associated with reactance and negatively associated with resistance and is considered a useful indicator of nutritional status. Malnutrition, with a characteristic water shift from intracellular to extracellular, is reflected in the phase angle [15]. The association between low phase-angle scores and postoperative outcomes and complications has been reported in cardiac patients and in patients with gynecologic cancer [16,17], as patients with lower phase-angle scores had longer hospitalization time.

The aim of the present study was to investigate body composition changes in the perioperative period in patients undergoing major head and neck surgery and to study postoperative cellular changes and their relation to perioperative fluid administration. Furthermore, we aimed to correlate different biochemical parameters and BIA with postoperative complication rate.

## 2. Materials and Methods

This is a prospective observational pilot study conducted between January 2019 and March 2020 at a single tertiary care center. The study protocol has been approved by the Rabin Medical Center Institutional Review Board (RMC-0255-18), with all participants signing a written consent form preoperatively. Patients without the ability to sign an informed consent were excluded. We included all eligible adult patients who underwent major head and neck surgery that required postoperative nasogastric tube feeding (oral cavity and oropharynx ablative surgery with or without reconstruction and total laryngectomy). As with our department protocol, all patients underwent preoperative nutritional assessment by a professional dietitian and were prescribed an oral supplement of immune-modulating nutrition (Impact, Nestle, Switzerland) for five consecutive days preoperatively as part of department protocol.

All patients were operated by a senior, specialist head and neck surgeon, and a senior anesthesiologist. Postoperatively, all patients were admitted to the otolaryngology—head and neck surgery ward after a recovery period in the recovery room. Nasogastric tube feeding was administered from postoperative day (POD) 1 at a continuous low rate, which escalated up to POD 4, achieving the full nutritional goals defined by predictive equations. Postoperative immunonutrition was continued until POD 5 only in patients compliant with their preoperative recommendations. Throughout the hospitalization period, a daily dietitian assessment and adjustments were performed by a professional dietitian according to body weight, serum albumin, pre-albumin, and white blood cells (WBCs) levels.

Patients’ demographic and clinical data were retrieved. Intraoperative and postoperative fluid administrations were recorded, as well as postoperative complications and hospital length of stay. Nutritional status was assessed at three different time points: preoperatively and at POD 2 and POD 10. Height and weight were measured for BMI calculation, full blood analysis was obtained, including serum albumin and pre-albumin levels, and body composition measures were performed using the Quadscan 4000 (Bodystat Ltd., Cronkbourne Douglas, Isle of Man, UK) multifrequency BIA apparatus. Impedance measurement was measured as resistance to flow of current between two electrodes placed on the wrist and the ankle. Phase angle (PA) is based on changes in resistance and reactance as alternating current passes through tissues, which causes a phase shift and is being calculated by the BIA. The value used was the 5th percentile according to the Bosy-Westphal study conducted on the German population and validated in patients with cancer by Norman et al. [18,19].

The Clavien–Dindo classification (CDC) was used to evaluate postoperative complications according to five different grades [20,21]. Grade 1 included minor risk events not requiring therapy. Grade 2 complications were defined as complications that require pharmacological intervention. Grade 3 complications were defined as complications that require surgical intervention. Grade 4 complications were defined as complications leading to permanent disability or organ lost, and grade 5 was defined as complications that lead to the death of a patient. Additionally, local complications were documented as wound dehiscence, abscess formation, orocutaneous or pharyngocutaneous fistula, hematoma, hemorrhage, seroma, and flap lost.

Statistical analysis was performed with SPSS software, version 21.0 (IBM Corp., Armonk, NY, USA). Categorical variables were described as frequency and percentage. Normally distributed variables were described as mean and standard deviation, and abnormally distributed variables as median and range. Categorical variables were compared with the chi-square test; normally distributed variables were compared with the *t*-test; and nonparametric variables compared with the Mann–Whitney test. Linear associations were assessed by Pearson correlation coefficient (PCC). Logistic regression was performed for multivariate analysis. A *p* value of <0.05 was considered significant.

## 3. Results

A total of 37 patients were included in our study. Three patients did not complete the postoperative nutritional measurements due to postoperative delirium and therefore were excluded from the analysis. Twenty (58.8%) patients were female with a mean age of 62 (range, 35–84) years. Mean preoperative BMI was 26 (range 15.7–43.2). Thirty-one (91.1%) patients were operated on due to head and neck cancer: the majority of whom (90.3%) had advanced stage disease. Three patients had severe osteoradionecrosis (ORN) that required surgical intervention. Seventeen patients (50%) had a history of smoking, and nine patients (26.4%) were diabetic. There were no patients with a history of congestive heart failure, and all patients had Clinical Frailty Scale of 1–3. Table 1 presents the demographics and baseline clinical characteristics of all patients.

Mean operative time was 463 (range, 140–720) min. Fifteen patients underwent free vascular flap reconstruction, and the remaining patients had local or regional flaps and primary closure. The mean total intraoperative fluid administration and recovery room fluid administration were 3682 mL (SD ± 1910) and 1802 mL (SD ± 1466), respectively with a mean intraoperative fluid administration per hour of 486 mL/h. All perioperative fluids were crystalloids. Mean operative and recovery-room urine output were 678 mL (SD ± 490) and 965 mL (SD ± 752), respectively. Mean total perioperative fluid balance was 3641 mL (SD ± 1989), and a total of 22 (64.7%) of patients received vasopressin during the perioperative period.

### 3.1. Perioperative Metabolic Changes

Table 2 presents the metabolic changes for all patients during the perioperative period. Preoperatively, only eight (23.5%) patients had low phase-angle scores below the 5th percentile. On POD 2, all patients had a decrease in serum albumin and pre-albumin levels, higher extracellular water (ECW) percentile, and lower phase-angle scores. A total of 21 (61.7%) patients had a low phase-angle score below the 5th percentile. All the above parameters showed a trend toward their baseline at POD 10. Intracellular water (ICW) percentile did not change during the perioperative period.

Higher perioperative fluid administration was associated with POD-2 phase-angle score below the 5th percentile with a mean total volume of 6581 (SD ± 2626), compared to 3816 (SD ± 1778) in patients with a phase-angle score above the 5th percentile (*p* = 0.003). Total perioperative fluid administration was correlated with postoperative weight gain (r value 0.688, *p* = 0.012), excess in ECW percentage (Pearson correlation coefficient r value 0.731, *p* = 0.038), lower postoperative serum albumin and pre-albumin levels (PCC r values 0.611 and 0.480, *p* < 0.001 and *p* < 0.005, respectively) and lower phase-angle scores (PCC r value 0.851, *p* < 0.005).

Table 3 presents the distribution of perioperative fluids data according to the type of reconstruction. We compared patients that underwent free-flap reconstruction to non-free-flap reconstruction (regional flap or primary closure). Patients who underwent free-flap reconstruction received higher volume of fluids intraoperatively and in the recovery room (*p* <0.005), gained more weight in the perioperative period (*p* = 0.02), and had a lower phase-angle score at POD 2 (*p* = 0.032). Vasopressin administration was not different with reconstruction type (*p* = 0.832).

### 3.2. Metabolic Changes and Postoperative Complications

Sixteen patients experienced systemic or local postoperative complications for a total of 19 complications. Table 4 presents the distribution of all complications with reference to the type of reconstruction, and Table 5 presents the distribution of all complications with reference to the phase-angle scores. A preoperative phase-angle score below the 5th percentile was associated with systemic complications (*p* = 0.01), and a POD-2 phase-angle score below the 5th percentile was associated with a local as well as total complication rate (*p* = 0.035). Vasopressin administration was not associated with surgical nor medical complications (*p* = 0.959, *p* = 0.403 respectively). Smoking status and diabetes were also not associated with postoperative complications (*p* = 0.585 and *p* = 0.177, respectively).

We performed a multivariate analysis in order to try and investigate the independent impact of perioperative fluids and phase-angle score on postoperative complications. We found that when controlling for gender, sex, diabetes, smoking status, and free-flap reconstruction, high-volume perioperative fluids administration is not an independent risk factor for complications (Table 6).

Mean hospitalization time was 20.46 days (SD ± 13.45). There was no statistically significant difference in hospitalization length of stay with regard to reconstruction method (22.3 (SD ± 13.71) vs. 19.31 (SD ± 13.60)) (*p* = 0.153). However, patients with POD-2 phase-angle scores below the 5th percentile had statistically significant longer hospitalization stay (25.38 (SD ± 14.05) vs. 15.42 (SD ± 11.75)) (*p* = 0.029).

## 4. Discussion

In the present study, we investigated perioperative metabolic changes caused by perioperative volume administration and their association with postoperative complications. To our knowledge, this is the first study that examined bioelectrical impedance and phase-angle score and their relation to postoperative complications in head and neck patients. We have shown that perioperative fluid administration was strongly correlated with weight gain, ECW fluid accumulation, and a decrease in serum albumin and pre-albumin levels. Moreover, perioperative fluid administration correlated with low phase-angle score below the 5th percentile, which is a known marker for a low nutritional status. Poor preoperative nutritional status, measured by phase angle, was associated with postoperative systemic complication, while postoperative nutritional status was associated with local complications and longer hospitalization time.

Several studies have examined the relation between intraoperative fluid administration and postoperative complications in head and neck surgery. The association of medical or systemic complications and fluid administration is still a matter of debate. Studies by Farwell et al. and Patel et al. did not show correlations between intraoperative fluid administration and medical complication rates [22,23]. In a recent study by Dooley et al. on 102 head and neck patients who underwent vascular free-flap reconstruction, greater perioperative fluid administration was not found to be associated with postoperative medical complications. By contrast, Haughey et al. and Clark et al. showed a positive correlation between intraoperative fluid administration, greater than 130 mL/24 h or 7000 mL, respectively, and postoperative systemic complications [24,25].

The mechanism of injury regarding operative fluid administration on surgical complication in free-flap reconstruction was studied by Haughey et al. [24]. In their study, they found an association between intraoperative fluid administration greater than 7000 mL with flap complications. They suggested that mechanical stress due to recipient site edema causes flap failure. Similarly, Farwell et al. [22] reported that patients with postoperative surgical complication had higher operative fluid administration with a median of 6600 mL compared to 3400 mL. Dooley et al. [12] showed that not only intraoperative fluid administration is associated with surgical complications, but also the total fluid-administration volume, calculated as fluids administered in the operating room combined with fluids administered in the PACU. These were associated with postoperative surgical complications and therefore should be given with caution. In order to elucidate on the effects of perioperative fluid administration on cellular processes, we studied the postoperative metabolic changes on POD 2 and showed an excess of postoperative ECW. This can be explained by a fluid shift from the intracellular to the extracellular space or an intravascular circulation leakage from into the interstitial space due to a serum colloid pressure decrease as described by Shippy et al. [26]. An increased capillary permeability, resulting in fluid shifts from the vascular bed to the interstitial fluid to crystalloid infusions, may also contribute to secondary dilution. The fluids that accumulate in the extracellular space are associated with low phase-angle scores and are also associated with postoperative surgical complications and longer hospitalization time. It should be noted that postoperative surgical complications were not restricted to free-flap reconstruction only but were also documented in all patients who underwent major head and neck surgery. This association between low phase-angle scores and postoperative outcomes and complications has been reported in cardiac patients and in patients with gynecologic cancer, [16,17] as patients with lower phase-angle scores had longer hospitalization time. It should be noted that multivariate analysis failed to demonstrate that high-volume fluid administration and phase angle were independent factors for postoperative complications. This may be attributed to the correlation of total perioperative fluids, postoperative phase-angle score and diabetes and low sample size. The association between volume overload and diabetes has been studied in hemodialysis patients and in patients with chronic kidney disease. In this patient group, diabetic patients were prone for volume overload. In a study by Yildirim et al. [27], volume overload was more common in diabetic end-stage chronic kidney disease than in nondiabetic patients. They stated that volume overload and diabetes mellitus both cause impairment in endothelial function. In another study by Hung et al. that examined patients with chronic kidney disease, volume overload measured by a bioimpedance device was an independent risk factor for diabetes and cardiovascular disease [28].

The relationship between postoperative hypoalbuminemia and complications has been previously described. In a study by Tsai et al. [29] that analyzed 223 patients with head and neck vascular free-flap reconstruction, patients with postoperative serum albumin level of less than 3.5 mg/dL had a higher rate of major postoperative wound infection, defined as a postoperative recipient-site wound condition that required wound debridement in the operating room. It should be emphasized that other medical or surgical complications were not evaluated in that study. Hoppe et al. [30] in their study of 107 patients with head and neck free tissue transfer found that a postoperative serum albumin level of less than 2.5 mg/dL was associated with higher rates of surgical complications. Similarly, in our study, high perioperative fluid administration correlated with low mean postoperative serum albumin and pre-albumin levels of 3 mg/dL and 14 mg/dL, respectively. This can be explained by a fluid shift that couples with a decrease in serum albumin and is also associated with a decrease in other vital serum proteins of the immune system, which are essential in preventing wound infection.

Perioperative goal-directed therapy (GDT) refers to the hemodynamic optimization during perioperative care by titrating fluids, vasopressors, and/or inotropes to predefined hemodynamic goals. A meta-analysis by Hamilton et al. [31] has shown that GDT can reduce postoperative mortality and morbidity in moderate and high-risk surgical patients. Despite this, perioperative GDT is not carried out widely. Interestingly, in the head and neck field, there are no data regarding perioperative GDT [32]. Moreover, although a vast number of randomized trials have examined the effect of perioperative GDT, a recent meta-analysis concluded that no uniform conclusion on the effect of perioperative goal-directed therapy can be made due to a vast amount of clinical heterogeneity.

The present study has several limitations. First, as this was a prospective observational study, it consists of a relatively small sample size. Nevertheless, by creating a nutritional homogenic group with similar postoperative nasogastric tube feeding patterns, this study can propose a scientific explanation to a well-established clinical observation.

## 5. Conclusions

We conclude that the administration of high-volume perioperative fluids may affect postoperative nutritional status in head and neck patients, by causing fluids to shift toward the extracellular space. Furthermore, high-volume fluid administration is associated with several comorbidities, which may lead to postoperative complications and longer hospital length of stay in patients undergoing major head and neck surgery. Adjusted, low-volume perioperative fluid regimen should be considered in this patient group in order to minimize postoperative morbidity. Future randomized trials on perioperative goal-directed therapy in head and neck patients are warranted and may guide hemodynamic goals in this patient group.

## Figures and Tables

**Table 1 jcm-10-00539-t001:** Demographic and clinical characteristics of the cohort.

Parameter	Number (%)
Gender	
Male	14 (41.2%)
Female	20 (58.8%)
Mean age	62 (SD ±12.6)
Smoking history	
None	17 (50%)
Diabetes	9 (26.4%)
Reconstruction	
Primary closure	13 (38.3%)
Regional flap	6 (17.6%)
Free flap	15 (44.1%)
O Radial forearm	5 (33%)
O ALT ^1^	5 (33%)
O Fibula	5 (33%)

^1^ ALT—anterior lateral thigh.

**Table 2 jcm-10-00539-t002:** Metabolic changes of all patients during the perioperative period.

Parameter	Preoperatively	POD 2	POD 10
BMI	26 (±6.73)	25.8 (±6.23)	23.06 (±4.9)
Albumin	4.17 (±0.421)	3.01 (±0.65)	3.57 (±0.64)
Pre-albumin	22.27 (±5.58)	13.91 (±5.86)	19.4 (±4.93)
ECW%	24.1 (±4.08)	28.39 (±6.41)	24.8 (±6.43)
ECW (Lt.)	17.08 (±3.43)	20.1 (±4.35)	17.08 (±4.84)
ICW%	29.87 (±5.3)	29.77 (±6.18)	31.69 (±7.13)
Phase angle	6.35 (±1.62)	4.96 (±1.6)	5.93 (±1.74)

BMI, body mass index; ECW, extracellular weight; ICW, intracellular weight; POD, postoperative day.

**Table 3 jcm-10-00539-t003:** Distribution of perioperative fluids data according to reconstruction.

Parameter	Free-Flap Reconstruction (*n* = 15)	Regional Flap or No Reconstruction (*n* = 19)	*p* Value
Operative fluids administration (mL)	4773 (±2026)	2821 (±1312)	<0.005
Recovery room fluids administration (mL)	2556 (±1482)	1200 (±1268)	<0.005
Total perioperative fluids administration (mL)	7340 (±2257)	4021 (±2048)	<0.001
Total perioperative fluid balance * (mL)	5157 (±1871)	2802 (±1398)	<0.001
Operative vasopressin administration	7 (46.6%)	8 (42.1%)	0.832
Perioperative weight difference ** (Lt.)	+3.01 (±5.91)	−2.5 (±6.45)	0.02
Postoperative ECW difference (Lt.)	+4.33 (±3.88)	+2.08 (±3.94)	0.083
Preoperative phase angle (°)	5.92 (±1.54)	6.67 (±1.5)	0.298
POD-2 phase angle (°)	4.32 (±1.12)	5.49 (±1.77)	0.032

* Total perioperative fluid balance was measured as the difference between total perioperative fluids administration and total perioperative urine output. **** Perioperative weight difference was measured as the difference between the preoperative weight to POD-2 weight. POD, postoperative day.

**Table 4 jcm-10-00539-t004:** Distribution of all complications with reference to the type of reconstruction.

Parameter		All Patients Number (%)	Free-Flap Reconstruction (*n* = 15)	Other or No Reconstruction (*n* = 19)	*p* Value
Systemic complications *	1	-	-	-	0.229
	2	1 (2.9%)	-	1	
	3	5 (14.7%)	4	1	
	4	1 (2.9%)	-	1	
	5	-	-	-	
Local complications	Infection	5 (2.9%)	3	2	
	Hematoma	-	-	-	
	Dehiscence	1 (2.9%)	1	-	0.047
	Fistula formation	3 (8.8%)	3	-	
	Flap failure	3 (8.8%)	3	-	
Total complications *	1	3 (8.8%)	1	2	
	2	6 (17.6%)	3	3	
	3	6 (17.6%)	5	1	0.411
	4	4 (11.7%)	3	1	
	5	-	-	-	

* According to the Clavien–Dindo classification (CDC).

**Table 5 jcm-10-00539-t005:** Distribution of all complications with reference to phase-angle score.

Parameter		Preoperative Phase Angle		*p* Value	POD-2 Phase Angle		*p* Value
		Above 5 (*n* = 25) Below 5 (*n* = 9)			Above 5 (*n* = 13) Below 5 (*n* = 21)		
Systemic complications *****	1	-	1		3	1	
	2	1	3		2	-	
	3	1	1	0.01	1	-	0.134
	4		-		-	-	
	5	-	-		-	-	
Local complications	Infection	4	1		-	5	
	Hematoma	-	-		-	-	
	Dehiscence	-	1	0.44	-	1	0.035
	Fistula formation	2	1		2	1	
	Flap failure	2	1		-	3	
Total complications *****	1	-	1		5	1	
	2	5	5		2	6	
	3	3	2	0.933	1	-	0.035
	4	2	1		-	4	
	5	-	-		-	-	

* According to the Clavien-Dindo classification (CDC).

**Table 6 jcm-10-00539-t006:** Multivariate logistic regression analysis of total complications.

Parameter	Assessment Coefficient	SE	Wald	*p* Value	OR (95% CI)
Age	−0.085	0.064	1.749	0.186	0.919 (0.811–1.042)
Sex	−0.170	1.243	0.019	0.891	0.844 (0.074–9.646)
Smoking status	0.021	1.427	0.703	0.402	0.204 (0.008–6.988)
Diabetes	2.667	1.323	4.067	0.044	14.403 (1.078–192.424)
Free-flap reconstruction	−0.757	1.353	0.313	0.576	0.469 (0.033–6.653
Total perioperative fluids	−0.685	0.337	4.127	0.042	0.504 (0.260–0.976)
POD-2 phase angle	0.050	0.446	0.013	0.910	1.052 (0.439–0.2521)

## Data Availability

The data presented in this study are available on request from the corresponding author. The data are not publicly available due to privacy reasons.

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
