# Peer review of "Bioelectrical Impedance Analysis in Patients Undergoing Major Head and Neck Surgery: A Prospective Observational Pilot Study"

_jcm, 2021, doi:10.3390/jcm10030539_

Round 1

Reviewer 1 Report

Review: Bioelectrical impedance analysis in patients undergoing major 2 head and neck surgery: a prospective observational study

This is a prospective study which aims to evaluate perioperative body composition and cellular changes that occur in patients undergoing head and neck surgery and whether these changes affect complication rates. The findings are that increased perioperative fluid administration is associated with high extra-cellular water percentages and low scores by bio-impedance analysis (BIA). Such patients had higher complication rates and longer hospital stays. The manuscript is generally well-written and the study is warranted as there continues to be wide-spread issues with fluid overload of patients in the perioperative period. However, there are some important pieces of information that are missing and I have specific questions and comments, both major and minor, which I have included below.

Major points:

  1. In the methods the determination of BIA is briefly stated with little detail. I think it would be difficult for a reader with no expertise in this area to understand how the BIA was measured and what “phase angle” measures and how exactly it is determined.
  2. The mean metabolic values for patients in the preoperative period is stated in Table 2, though there is no detail of preoperative nutritional status in general. It seems the albumins/pre-albumins were fairly normal. Does this mean that most/all the patients had normal nutritional status prior to surgery, despite head and neck disease?
  3. Did the hospital/service have any policies in place for judicious perioperative fluid administration?
  4. Some of the conclusions seem obvious..that increased fluids lead to increased extracellular fluids and hypoalbuminemia. The point of interest about BIA is a little lost in the discussion/explanations in the background/methods and discussion sections.

Minor points:

  1. The abstract mentions the results of “low phase angle” without explaining what this phrase means. I think it would be difficult for readers to understand the concept by reading the abstract.
  2. The number of different surgeons and anesthesiologists involved in the procedures would be useful to know. Were these all specialists, particular the anesthesia staff? Did they have similar practices?
  3. What were the reasons for large percentages of patients to require vasopressin?
  4. The authors put forward some possible explanation for flap failure with higher perioperative fluid administration. What are their thoughts about the higher rates of infection?
  5. There are some minor grammatical errors and some misspellings e.g. line 76…should this read “predictive equations”?

Author Response

Dear Reviewer,

Please find attached revisions for the manuscript entitled “Bioelectrical impedance analysis in patients undergoing major head and neck surgery: a prospective observational study”. We appreciate the editorial board review and hope these revisions address relevant concerns. Revisions are highlighted throughout the manuscript in yellow.

Thank you for your time and consideration. Please do not hesitate to contact us with any questions or concerns.

Sincerely,

Sharon Tzelnick, MD, MPH

Notes:

Editor and Referee comments are in bold italic.

Responses from the authors are in regular.

Line numbers cited by reviewers refer to the original manuscript.

Line numbers cited in the authors' response refer to the revised manuscript.

Changes in the revised manuscript are highlighted.

Reviewer #1

We thank the reviewer for taking the time to evaluate the manuscript and have revised it accordingly.  

Major points:

  1. In the methods the determination of BIA is briefly stated with little detail. I think it would be difficult for a reader with no expertise in this area to understand how the BIA was measured and what “phase angle” measures and how exactly it is determined.

Thank you for your comment. We have revised the Materials and Methods section accordingly.

Page 3, line 99-106:

“body composition measures performed using the Quadscan 4000 (Bodystat Ltd., Cronkbourne Douglas, Isle of Man, UK) multi-frequency BIA apparatus. Impedance measurement was measured as resistance to flow of current between two electrodes placed on the wrist and the ankle.

Phase angle (PA) is based on changes in resistance and reactance as alternating current passes through tissues, which causes a phase shift and is being calculated by the BIA. the value used was the 5th percentile according to the Bosy-Westphal study conducted on the German population and vali-dated in patients with cancer by Norman et al [18,19].”

  1. The mean metabolic values for patients in the preoperative period is stated in Table 2, though there is no detail of preoperative nutritional status in general. It seems the albumins/pre-albumins were fairly normal. Does this mean that most/all the patients had normal nutritional status prior to surgery, despite head and neck disease?

Thank you for your comment. Mean pre-operative BMI in our cohort was 26 (±6.73). This correlates with the mean normal albumins/pre-albumins. Only five patients (14.7%) were defined as pre-operatively malnourished with BMI<18.5%.

  1. Did the hospital/service have any policies in place for judicious perioperative fluid administration?

Thank you for your comment. At the moment we are revising our operative protocol of major head and neck surgeries by designating multidisciplinary team that include anesthesiologists and nursing.   

  1. Some of the conclusions seem obvious. that increased fluids lead to increased extracellular fluids and hypoalbuminemia. The point of interest about BIA is a little lost in the discussion/explanations in the background/methods and discussion sections.

We appreciate the reviewer’s comment and revised our study accordingly –

 Introduction

 page 2, lines 55-63:

“Nutrition can also be expressed as body composition. In the last two decades, bio-electrical impedance analysis (BIA) has been used to estimate body composition by measuring body component resistance (R) and reactance (Xc) by recording a voltage drop in applied current. Resistance related to the amount of water present in the tissues and reactance causes the current to lag behind the voltage creating a phase shift [14]. One of the most clinically established impedance parameters in BIA is the phase angle. Phase angle, a marker of soft tissue mass and hydration status, is positively associated with reactance and negatively associated with resistance, and is considered a useful indicator of nutritional status.”

Materials and Methods

Page 3, line 99-106:

“body composition measures performed using the Quadscan 4000 (Bodystat Ltd., Cronkbourne Douglas, Isle of Man, UK) multi-frequency BIA apparatus. Impedance measurement was measured as resistance to flow of current between two electrodes placed on the wrist and the ankle.

Phase angle (PA) is based on changes in resistance and reactance as alternating current passes through tissues, which causes a phase shift and is being calculated by the BIA. The value used was the 5th percentile according to the Bosy-Westphal study conducted on the German population and vali-dated in patients with cancer by Norman et al [18,19].”

Minor points:

  1. The abstract mentions the results of “low phase angle” without explaining what this phrase means. I think it would be difficult for readers to understand the concept by reading the abstract.

Thank you for your comment. We have revised the Abstract accordingly.

Page 1, line 23-25:

“Total perioperative fluid administration was associated with post-operative high extra-cellular water percentages (P=0.038) and low phase angle score (P <0.005) which indicates low nutritional status.”

  1. The number of different surgeons and anesthesiologists involved in the procedures would be useful to know. Were these all specialists, particular the anesthesia staff? Did they have similar practices?

Thank you for your comment. In the study period, all patients underwent major head and neck surgery were operated by a senior, specialist head and neck surgeon and a senior anesthesiologist.

We have revised the Materials and Methods section accordingly –

Page 2, line 85-86:

“All patients were operated by a senior, specialist head and neck surgeon and a senior anesthesiologist.”

  1. What were the reasons for large percentages of patients to require vasopressin?

Thank you for your comment. Hemodynamic stability is determined according to anesthesiologist preference and in our institution the use of vasopressin is acceptable in order to decrease volume overload in the major head and neck surgeries. In a recent study that was published in the Annals of Surgery (Fang L. et al, Intraoperative Use of Vasopressors Does Not Increase the Risk of Free Flap Compromise and Failure in Cancer Patients. Ann Surg. 2018 Aug;268(2):379-384), vasopressin was not associated with higher risk for free flap failure.

  1. The authors put forward some possible explanation for flap failure with higher perioperative fluid administration. What are their thoughts about the higher rates of infection?

Thank you for your comment. We believe that in accordance to flap failure, post-operative infection is also associate with fluid overload and nutritional status. In table 5 we present the association between low phase angle score and operative complication, which includes post-operative infection. This is also discussed in the Discussion section in page 8, lines 255-268.

  1. There are some minor grammatical errors and some misspellings e.g. line 76…should this read “predictive equations”?

Thank you for your comment. I did not find the grammatical errors in line 76. If you would be able to point out again for the misspellings so I can correct them.

Thank you for your kind consideration of this manuscript. We hope the changes adequately address the reviewers’ comments and meet with your approval. All authors have seen and agreed to the changes as submitted.

Reviewer 2 Report

The article ''Bioelectrical impedance analysis in patients undergoing major 2 head and neck surgery: a prospective observational study'' is a well written article by Gideon Bachar group.The authors investigated the perioperative changes in patients undergoing major head and neck surgery and to examine the impact of perioperative fluid administration on body composition and metabolic changes using bioelectrical impedance which is very novel and interesting. The authors also conclude that High-volume perioperative fluids administration impacts postoperative nutritional status with fluid shift towards the extra-cellular space, associated with higher postoperative complication rate and longer LOS

Overall the article looks very interesting and could be a good fit for the journal. Hence, I would suggest acceptance without any major changes.

Author Response

Thank you for your time and kind consideration of this manuscript.

Reviewer 3 Report

The aim of this work is to study perioperative changes in 16 patients undergoing major head and neck surgery and to examine the impact of perioperative fluid administration on body composition and metabolic changes through bioelectric impedance.

The work presents a good hypothesis, makes clinical and scientific sense and biological plausibility, although it is true that there are too many objectives for such a low sample size. The introduction is well written and organized and no plagiarism or self-citations have been detected. The material and methods are well developed and allow the reproducibility of the study, although the reference of the ethics committee must be provided.

Among the main limitations of this study is the sample size, associated with a wide range of therapeutic varieties "head and neck patients", which perhaps for a N <40, makes the extrapolation of results difficult. Some treatments: “O Radial forearm, O ALT, O Fibula, they only include 5 patients so this is more in line with a pilot study. It could also increase power and reduce bias, reducing study groups. In any of the cases, no sample size calculation and / or statistical power associated with it is provided.

The results are generally well expressed with correct subsections, but it should be noted that the correlation coefficients should be indicated, not only the p-value, for the bivariate correlation study.

I am not very convinced that the free flap Vs other (or no reconstruction) groups are equally comparable groups. As group 2 others, it includes a wide variety of invasive treatments as well as untreated patients. Perhaps consider three groups, one of them being No reconstruction, it is possible that the differences between groups disappear or are strengthened in the group of more invasive treatments.

Regarding the objective “distribution of all complications with reference to the phase angle score”, due to the small sample size, statistical significance should be considered with caution.

It would be necessary, due to the high number of objectives and variables collected, to perform a multivariate logistic regression analysis, at least to determine the role of the phase angle in the evolution parameters, and thus obtain less biased results.

The conclusions are too ambitious for such a small sample size and must be reformulated.

Author Response

Dear Reviewer,

Please find attached revisions for the manuscript entitled “Bioelectrical impedance analysis in patients undergoing major head and neck surgery: a prospective observational study”. We appreciate the editorial board review and hope these revisions address relevant concerns. Revisions are highlighted throughout the manuscript in yellow.

Thank you for your time and consideration. Please do not hesitate to contact us with any questions or concerns.

Sincerely,

Sharon Tzelnick, MD, MPH

Notes:

Editor and Referee comments are in bold italic.

Responses from the authors are in regular.

Line numbers cited by reviewers refer to the original manuscript.

Line numbers cited in the authors' response refer to the revised manuscript.

Changes in the revised manuscript are highlighted.

Reviewer #3

We thank the reviewer for taking the time to evaluate the manuscript and have revised it accordingly.  

  1. The aim of this work is to study perioperative changes in 16 patients undergoing major head and neck surgery and to examine the impact of perioperative fluid administration on body composition and metabolic changes through bioelectric impedance.

The work presents a good hypothesis, makes clinical and scientific sense and biological plausibility, although it is true that there are too many objectives for such a low sample size. The introduction is well written and organized and no plagiarism or self-citations have been detected. The material and methods are well developed and allow the reproducibility of the study, although the reference of the ethics committee must be provided.

Thank you for your comment. In the material and method section, page 2, lines 75-77, we have provided the data regarding the ethics committee – “The study protocol has been approved by the Rabin Medical Center Institutional Review Board, with all participants signing a written consent form pre-operatively”

  1. Among the main limitations of this study is the sample size, associated with a wide range of therapeutic varieties "head and neck patients", which perhaps for a N <40, makes the extrapolation of results difficult. Some treatments: “O Radial forearm, O ALT, O Fibula, they only include 5 patients so this is more in line with a pilot study. It could also increase power and reduce bias, reducing study groups. In any of the cases, no sample size calculation and / or statistical power associated with it is provided.

Thank you for your comment. We agree with the reviewer that due to the low sample size it was difficult to create sub analyses regarding different reconstruction types. Therefore, in our manuscript we included all free flap reconstruction as one group.

  1. The results are generally well expressed with correct subsections, but it should be noted that the correlation coefficients should be indicated, not only the p-value, for the bivariate correlation study.

Thank you for your comment. We have revised the text accordingly and elaborate on the correlation coefficients for the bivariate correlation analysis.

Page 5, lines 168-173:

“Total perioperative fluid administration was correlated with post-operative weight gain (r value 0.688, P=0.012), excess in ECW percentage (Pearson correlation coefficient r value 0.731, P=0.038), lower post-operative serum albumin and pre-albumin levels (PCC r values 0.611 and 0.480, P <0.001 and P <0.005, respectively) and lower phase angle score (PCC r value 0.851, P <0.005).”

  1. I am not very convinced that the free flap Vs other (or no reconstruction) groups are equally comparable groups. As group 2 others, it includes a wide variety of invasive treatments as well as untreated patients. Perhaps consider three groups, one of them being No reconstruction, it is possible that the differences between groups disappear or are strengthened in the group of more invasive treatments.

Thank you for your comment. All patients in our study were operated. In table 3 we compared patients that underwent free-flap reconstruction to non-free flap reconstruction (regional flap or primary closure). Our decision to consider regional flap and non-reconstructed patients into one group stemmed from the similarity of both groups regarding operative time and fluid managements compare to patients undergoing free flap reconstructions.

We have revised the Results section and table 3 accordingly.

  1. Regarding the objective “distribution of all complications with reference to the phase angle score”, due to the small sample size, statistical significance should be considered with caution. It would be necessary, due to the high number of objectives and variables collected, to perform a multivariate logistic regression analysis, at least to determine the role of the phase angle in the evolution parameters, and thus obtain less biased results.

Thank you for your comment. We revised the text accordingly and perform a multivariate logistic regression analysis.

Page 7, lines 201-205:

“We performed a multi-variate analysis in order to try and investigate the independent impact of perioperative fluids and phase angle score on post-operative complications. We found that when controlling for gender, sex, diabetes, smoking status and free flap reconstruction, high-volume perioperative administration was an independent risk factor for complications (P-value 0.042). [Table 6]”

Parameter

Assessment coefficient

SE

Wald

P value

OR (95% CI)

Age

-0.085

0.064

1.749

0.186

0.919 (0.811-1.042)

Sex

-0.170

1.243

0.019

0.891

0.844 (0.074-9.646)

Smoking status

0.021

1.427

0.703

0.402

0.204 (0.008-6.988)

Diabetes

2.667

1.323

4.067

0.044

14.403 (1.078-192.424)

Free flap         reconstruction

-0.757

1.353

0.313

0.576

0.469 (0.033-6.653

Total perioperative fluids

-0.685

0.337

4.127

0.042

0.504 (0.260-0.976)

POD2 phase angle

0.050

0.446

0.013

0.910

1.052 (0.439-0.2521)

able 6. multivariate logistic regression analysis of total complications

Page 8, lines 261-265”

“A multivariate analysis demonstrated that high-volume fluid administration was an independent factor and for post-operative complications. We could not show a similar correlation regarding phase angle score in POD 2. This may be attributed to the correlation of total per-operative fluids and post-operative phase angle score and low sample size.”

  1. The conclusions are too ambitious for such a small sample size and must be reformulated.

We appreciate your comment and revised our conclusion accordingly.

Page 8, lines 284-291:

We conclude that administration of high-volume perioperative fluids affects postoperative nutritional status in Head and Neck patients, by causing fluids to shift towards the extra-cellular space. It may be associated with higher post-operative com-plication rates and longer hospital lengths of stay in patients undergoing major head and neck surgery. Adjusted, low-volume perioperative fluid regimen should be considered in this patients group in order to minimize postoperative morbidity. Future randomize trials on perioperative goal‐directed therapy in Head and Neck patients is warranted and may guide hemodynamic goals in this patient group.

Thank you for your kind consideration of this manuscript. We hope the changes adequately address the reviewers’ comments and meet with your approval. All authors have seen and agreed to the changes as submitted.

Round 2

Reviewer 1 Report

Thank you to the authors for their responses and updates to the manuscript. The explanations of their methods and findings are much clearer.

Author Response

Thank you for your kind consideration of this manuscript

Reviewer 3 Report

- An appropriate Reference number for the ethical committee must be provided.

- Title must be reformulated according to this pilot study

- Abstract must be changed according to the new data obtained by the multivariate regression analysis.

- I honestly believe that the regression analysis is not correctly interpreted, as far as according to the data, the OR for the total perioperative fluids was 0.504 (0.260-0.976), which suggests a protective (not risk) effect. On the other hand, it looks that the presence of diabetes is more related to the post-operative complications: OR 14.403 (1.078-192.424); p=0.044. Please reconsider these results, express and discuss it accordingly.

Conclusions are still not supported by the results.

Thank you.

Author Response

We thank the reviewer for taking the time to evaluate the manuscript and have revised it accordingly.  

  1. An appropriate Reference number for the ethical committee must be provided.

Thank you for your comment. We have revised the text accordingly and added an appropriate reference number for the ethical committee.

Page 2, lines 76-77

“The study protocol has been approved by the Rabin Medical Center Institutional Review Board (RMC-0255-18), with all participants signing a written consent form pre-operatively.”

  1. Title must be reformulated according to this pilot study

Thank you for your comment. We have revised the title accordingly

“Bioelectrical impedance analysis in patients undergoing major head and neck surgery: a prospective observational pilot study”

  1. Abstract must be changed according to the new data obtained by the multivariate regression analysis.

Thank you for your comment. We have revised the Abstract section accordingly.

Page 1, lines 27-33:

"Multivariate analysis failed to demonstrate that high-volume fluid administration and phase angle are independent factors for post-operative complications.  High-volume perioperative fluids administration impacts postoperative nutritional status with fluid shift towards the extra cellular space and is associated with factors that increase the risk of post-operative complications and longer LOS. An adjusted, low-volume perioperative fluid regimen should be considered in patients with co-morbidities in order to minimize postoperative morbidity."

  1. I honestly believe that the regression analysis is not correctly interpreted, as far as according to the data, the OR for the total perioperative fluids was 0.504 (0.260-0.976), which suggests a protective (not risk) effect. On the other hand, it looks that the presence of diabetes is more related to the post-operative complications: OR 14.403 (1.078-192.424); p=0.044. Please reconsider these results, express and discuss it accordingly.

We appreciate you comment. We have revised the Results and Discussion accordingly.

Page 7, lines 212-216:

“We performed a multi-variate analysis in order to try and investigate the independent impact of perioperative fluids and phase angle score on post-operative complications. We found that when controlling for gender, sex, diabetes, smoking status and free flap reconstruction, we failed demonstrating that high-volume perioperative administration was an independent risk factor for complications. [Table 6]”

            Page 8, lines 272- 277:

“It should be noted that multivariate analysis failed to demonstrate that high-volume fluid administration and phase angle were independent factors for post-operative complications. This may be attributed to the correlation of total peri-operative fluids and post-operative phase angle score and low sample size.”

  1. Conclusions are still not supported by the results

Thank you for your comment. We have further elaborated on the relation of volume overload and comorbidities, especially diabetes in the Discussion section and Conclusion.

Page 8, lines 277-284:

“The association between volume overload and diabetes has been studied in hemodialysis patients and in patients with chronic kidney disease. In this patient group, diabetic patients were prone for volume overload. In a study by Yildirim et al. [27] volume overload was more common in diabetic end-stage chronic kidney disease than in non-diabetic patients. They stated that volume overload and diabetes mellitus both cause impairment in endothelial function. In another study by Hung et al. that examined patients with chronic kidney disease, volume overload measured by bioimpedance devise was an independent risk factor for diabetes and cardiovascular disease [28].”

  1. Yildirim Y, Kara AV, Kilinç F, Aydin F, E, Yilmaz Z, Kadiroglu AK, Yilmaz ME. Deter-mination of volume overload by bioelectrical impedance analysis and NT-PROBNP in diabetic pre-dialysis patients. Acta Endocrinol (Buchar). 2016;12(1):19-25
  2. Hung SC, Kuo KL, Peng CH, Wu CH, Lien YC, Wang YC, Tarng DC. Volume overload correlates with cardiovascular risk factors in patients with chronic kidney disease. Kidney Int. 2014 Mar;85(3):703-9

Thank you for your kind consideration of this manuscript. We hope the changes adequately address the reviewers’ comments and meet with your approval. All authors have seen and agreed to the changes as submitted.
